# Coupling analysis of contra-rotating fan interstage pressure pulsation and blade vibration based on wavelet reconstruction

Xinzhe Zhang[1,2,3], Chen Dai[2]*, Zunlong Jin[2], Sławomir Dykas[4], Guojie Zhang[2]*

1 School of Aeronautical Engineering, Zhengzhou University of Aeronautics, Zhengzhou, China, 2 School of Mechanical and Power Engineering, Zhengzhou University, Zhengzhou, China, 3 Censtar Science & Technology Corp., Ltd, Zhengzhou, China, 4 Department of Power Engineering and Turbomachinery, Silesian University of Technology, Gliwice, Poland

☯ These authors contributed equally to this work.
* dai_c_zzu@126.com (CD); zhangguojie2018@zzu.edu.cn (GZ)

**Data Availability Statement:** All relevant data are within the manuscript.

**Funding:** This study was funded by Science and Technology R&D Program of Henan Province of China to XZ (No.192102210056). The funders had

## Abstract

In recent years, the flow characteristics research of the interstage region in counter-rotating axial fans in terms of fluid dynamics has attracted much attention. Especially, the dynamic relationship between interstage pressure pulsation and blade vibration in counter-rotating axial fans has not yet been clarified. This paper performs the signal processing method of wavelet decomposition and reconstruction in time-frequency analysis process. Under different flow conditions, weak-coupling numerical simulation program is employed to analyze the fluid-structure coupling interaction between interstage pressure pulsations and blade vibrations in counter-rotating axial fans. The results indicate that the fluid-structure coupling interaction field in the interstage of counter-rotating axial fans mainly excites the first-order vibration of the second-stage blade. At the same time, the consistency between the pulsation frequency and the vibration frequency of the airflow reflects the good coupling property. Two stage blades cut the airflow to cause field changes and airflow pulsation, and then, airflow pulsation causes blades deformation and produces vibrations of the same frequency at the blade. The deformation of the blades, in turn, causes the flow field changes. Rotating stall, vortex movement and breakdown produced low-frequency airflow pulsation and vortex vibration of the blade.

## 1. Introduction

Rotation machinery, such as steam turbine, gas turbine, fan, compressor and so on, are employed in many industries [1, 2]. Counter-rotating axial fans, compared with general axial fans, can provide higher transmission efficiency, higher counter air volume and smaller structure size, which will become increasingly more important in axial fan develop process. However, the failure rate of counter-rotating axial fans is higher than that of general axial fans because of the particular nature of the structure, which is always the main source of decreased efficacy [3]. Therefore, it is very important to improve the reliability of counter-rotating axial

no role in study design, data collection and analysis, decision to publish, or preparation of the manuscript.

**Competing interests:** The authors have declared that no competing interests exist.

fans and optimize the design of the structure of spiral type axial flow ventilator. So, the fluid dynamics characteristics research receives a lot of attentions and becomes the current research focus [4]. Besides, the two-phase flow in counter-rotating axial fans should be paid more attentions [5, 6].

The nonlinear system of counter-rotating axial fans is complex, which is performed by the structure and the fluid [7]. He and Wang et al. [8, 9] researched the internal flow, vibration, noise about this type of the fan, especially, the design of the fan system, fault diagnosis, the structure improvement and so on. However, it is very little study on the influence of the inter-stage fluid flow in counter-rotating axial fans.

The main purpose of the time-frequency analysis is to study the frequency spectrum characteristics of signal along with changing time, and then to help us understand the meaning of time-varying frequency spectra in mathematics and physics. Finally, in the time and frequency, there is a lot of valuable characteristics information obtained through the various signal analysis [10]. Wavelet-based time-series analysis is also widely used in many different fields. Meantime, there are many researchers use this method to do a lot of work. Such as, PC Ivanov [11] used wavelet-based time-series to analyze scale behavior of heartbeat intervals, Mosabber Uddin Ahmed [12] used this method to analyze multivariate multiscale entropy and Zhong-Ke Gao [13–15] used the multiscale complex network to analyze experimental multivariate time series and so on. In addition, in the development of the signal processing technology used in time-frequency analysis, wavelet transforms become increasingly more important. The concept of wavelet transforms was first presented in 1984. And Takaaki Musha [16] confirmed the existence of a wavelet orthogonal basis of limited compact support, thus providing the famous Daubechies wavelet basis. Brazhe [17] devised the idea of multiscale analysis and summarized the Mallat algorithm for binary wavelet transforms as well as realized the engineering applications.

Wavelet transforms are characterized by their prominent localization ability and are very beneficial to detecting the transient state variations of signal [18]. To better identify weak signals and perform fault diagnosis for nonlinear systems, the signals also use continuous wavelet transforms according to the wavelet entropy to choose an optimal scale and then remove the extraneous signal interference using the method of wavelet reconstruction [19]. Based on the multi-resolution characteristics of signal analysis, the signals are subject to a detailed multi-scale analysis through the wavelet reconstruction method. We regarded the wavelet as a set of constantly transforming band-pass filters to filter the signals using the transform of the wavelet time-frequency window and then analyzed the interstage pressure pulsation signals of counter-rotating axial fans. We also summarized the fluid dynamics relationship between the fluid-structure coupling flow field interstage pressure pulsations and blade vibrations. Therefore, this paper performed the signal processing method of wavelet decomposition and reconstruction in time-frequency to analyze the specific flow field of counter-rotating axial fan interstage, then, under relative flow coefficients of $\Phi = 1$ and $\Phi = 0.605$, the fluid-structure interaction the counter-rotating axial fan interstage pressure pulsations and blade vibrations are analyzed.

## 2. Materials and methods

### 2.1 Experimental facilities

The experimental system consists of two parts including the contra-rotating fan experiment test part and signal measurement system (Fig 1). KDF-5 low noise contra-rotating fan is a core facility for the contra-rotating fan experiment test part. The parameters are shown in the follow (Table 1).

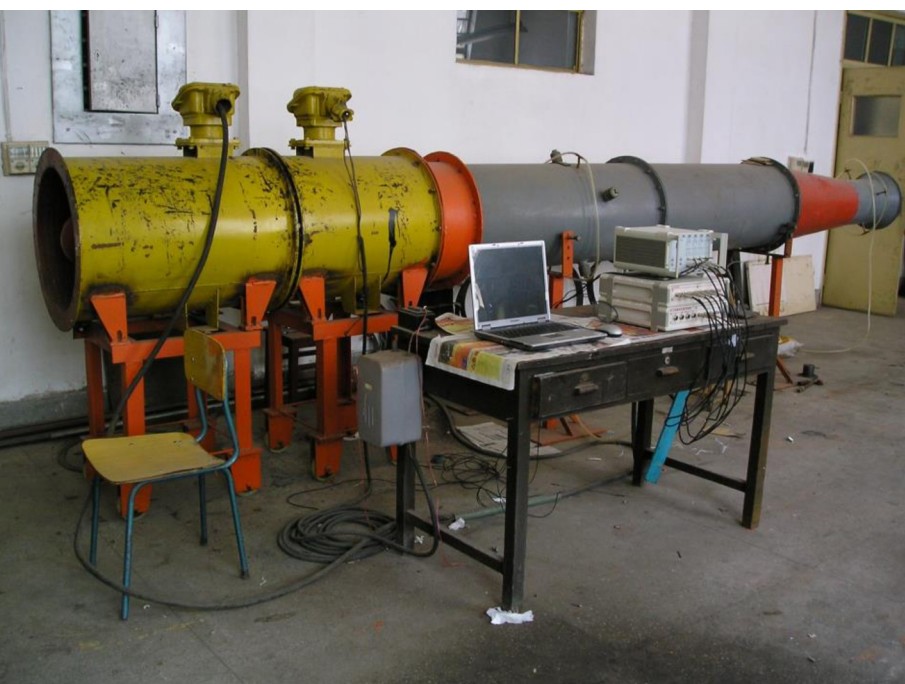

**Fig 1. KDF-5 low noise contra-rotating fan and main experimental facilities.**

The signal measurement system consists of sensors, microphone, amplifiers, ICP, data acquisition unit and data processor (Fig 2). The signal collection and conversion system mainly contain data acquisition unit, pre-regulator and pre-amplifier [20]. And it has the multi-functions of gain, smoothing, multiway switch, A/D convertor, memorizer and logic control circuit. The aerodynamic performance of the fan measured by u-shape tube manometer, vibration measured by KD1005 acceleration sensor and uT4304 multifunction pre-amplifier, airflow pulsation measured by CYB11 pressure sensor and pre-regulator, noise measured by AWA14425 microphone and AWA14604 pre-amplifier. Vibration, airflow pulsation and noise measurement are based on uTekL V2007 dynamic signal collection and analysis system, the output signal of the sensor is treated by pre-amplifier or pre-regulator through data acquisition unit to change the number signal.

## 2.2 Experimental methods

The direct measurement method is used in the process of signal acquisition. Then, the measurement of the blade vibration measured the impeller axial vibration. The blade vibration is

**Table 1. Parameters of KDF-5 fan.**

| Nomenclature | Parameters |
|---|---|
| Flow | 125~250m$^3$/min |
| Total pressure | 300~2800Pa |
| Rated power | 2×5.5kW |
| Rated speed | 2940rpm |
| Number of first-stage blades | 8 |
| Number of second-stage blades | 6 |
| Fan diameter | 0.51m |

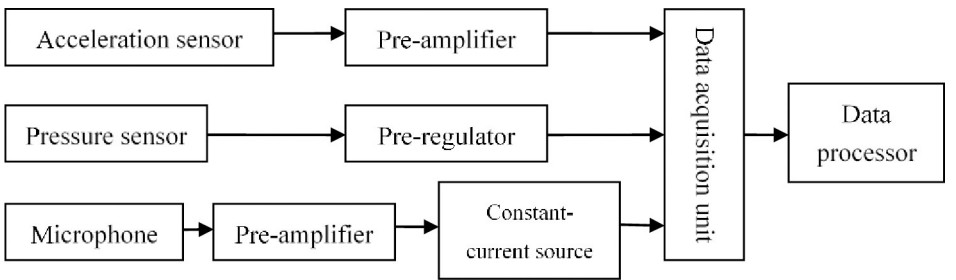

**Fig 2. Sketch of signal measurement system working principle.**

also reflected in the impeller axial vibration, due to the rigid connection of the blade and hub. Meantime, the fan axial vibration can reflect the whole flow field vibration and meet the airflow pulsation synchronization. For the airflow pulsation, it is measured by the pressure sensor in the interstage. According to Nyquist sampling theorem, we chose the 5120Hz sampling frequency and the 32 sampling data blocks. The sampling time is 6.4s. Measuring the fan axial vibration under different working conditions and the airflow pressure fluctuation in the corresponding interstage working condition, the signal measured by the acceleration sensor and the pressure sensor input the data acquisition system.

The experimental investigations on the interstage fluid flow were performed under five different relative flow coefficients (1, 0.770, 0.605, 0.479, 0.288), meantime, the corresponding numerical simulations analyses were compared. When relative flow coefficient was low 0.479, the result values exhibited a larger error (up5%) between numerical simulation and experimental measurement. So, it was also not suit to use the numerical simulation to analyze at the $\Phi = 0.288$ experiment condition. At $\Phi = 0.605$, the result values met the error range, the total pressure error was 4.31% and the efficiency error was 2.36%, at this condition, it was credible using the numerical simulation analysis. Then, at $\Phi = 0.605$, the amplitude of interstage pressure pulsation and blade vibrations obviously increased due to pressure mutations. Therefore, we chose the relative flow coefficients ($\Phi = 1$ and $\Phi = 0.605$) to make sure the follow scientific and targeted discussion results.

Base on the above experimental scheme, we chose five different radii (R = 170mm, R = 185mm, R = 200mm, R = 225mm, R = 245mm) analysis to the interstage airflow pressure pulsation at $\Phi = 1$ and $\Phi = 0.605$ respectively(Figs 3 and 4).

In the process of signal processing and analysis, the method of wavelet transform phase-space reconstruction [21] was applied. The key influence parameters of this method are time delay $\tau$ and embedding dimension $m$. The precision of time delay $\tau$ and embedding dimension $m$ is directly related with the accuracy of the invariables of the described characteristics of the strange attractors in phase space reconstruction [22]. The characteristics of the strange attractors of a chaotic system can be analyzed by sampling a part of the output chaotic time series of a system. The method that is commonly used is the state space reconstruction in delay coordinate proposed by Packard et al. [23]. It can be proved through Takens' theorem [24] that the unstable periodic obits (strange attractor) could be recovered properly in an embedding space whenever a suitable embedding dimension $m \geq 2d+1$ ($d$ is the dimension of chaotic system) is detected; that is, the obits in the reconstructed space $R^m$ keeps a differential homeomorphism with the original system. The algorithm that based on the AD method was used in this paper, by means of this algorithm, a near-optimum embedding dimension and delay time can be estimated.

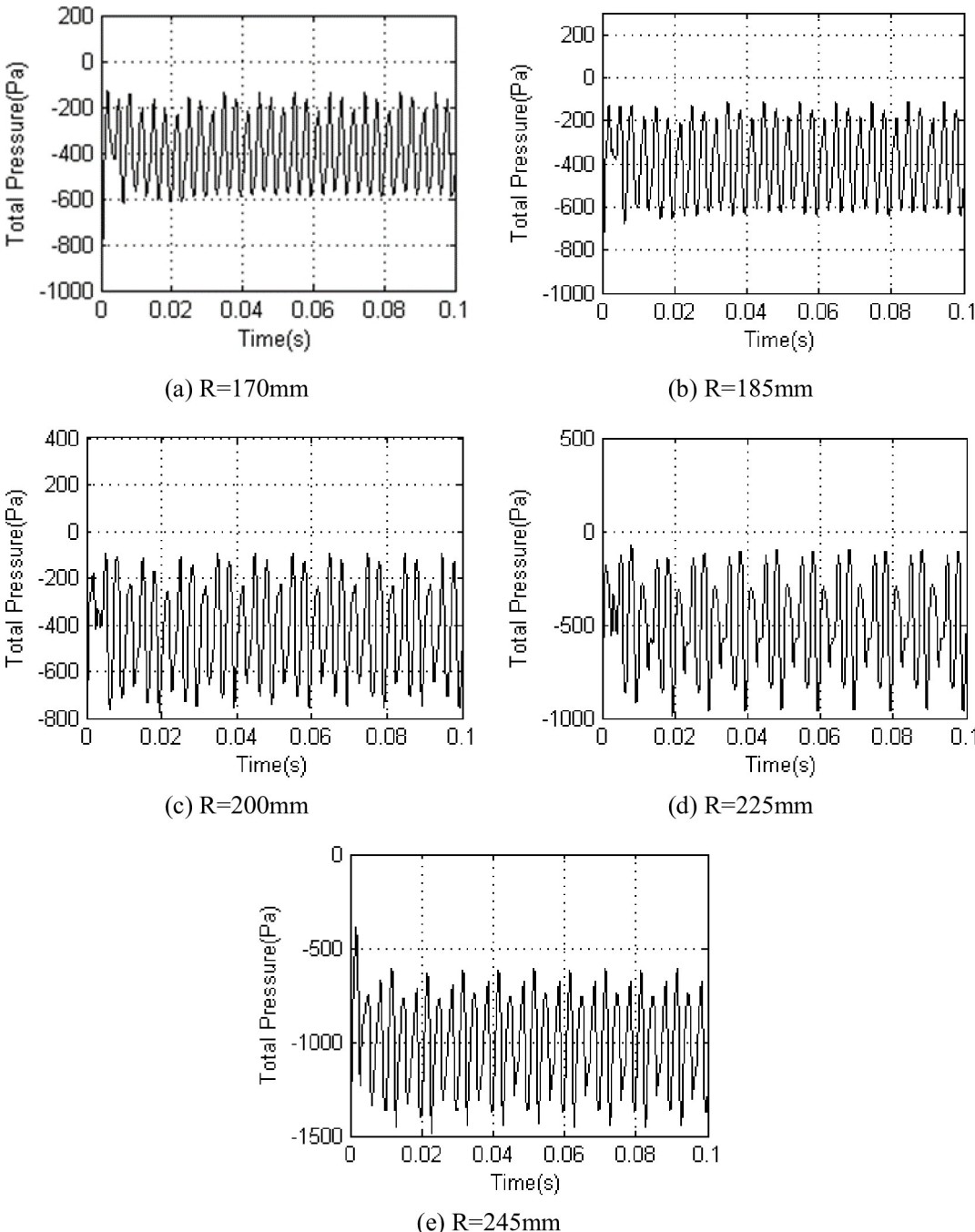

Fig 3. **Interstage pressure pulsation on different radius under** $\Phi = 1$.

## 3. Results and discussion

### 3.1 Experiment discussion

To better protect the numerical simulation results, we made the grid independence test for the fan model [25]. The computational grid had been generated for the calculation domain of the one blade passage using the algebraic generation method [26]. The grids had been tested to investigate the effect of grid refinement. Fine grid spacings were used near the leading and

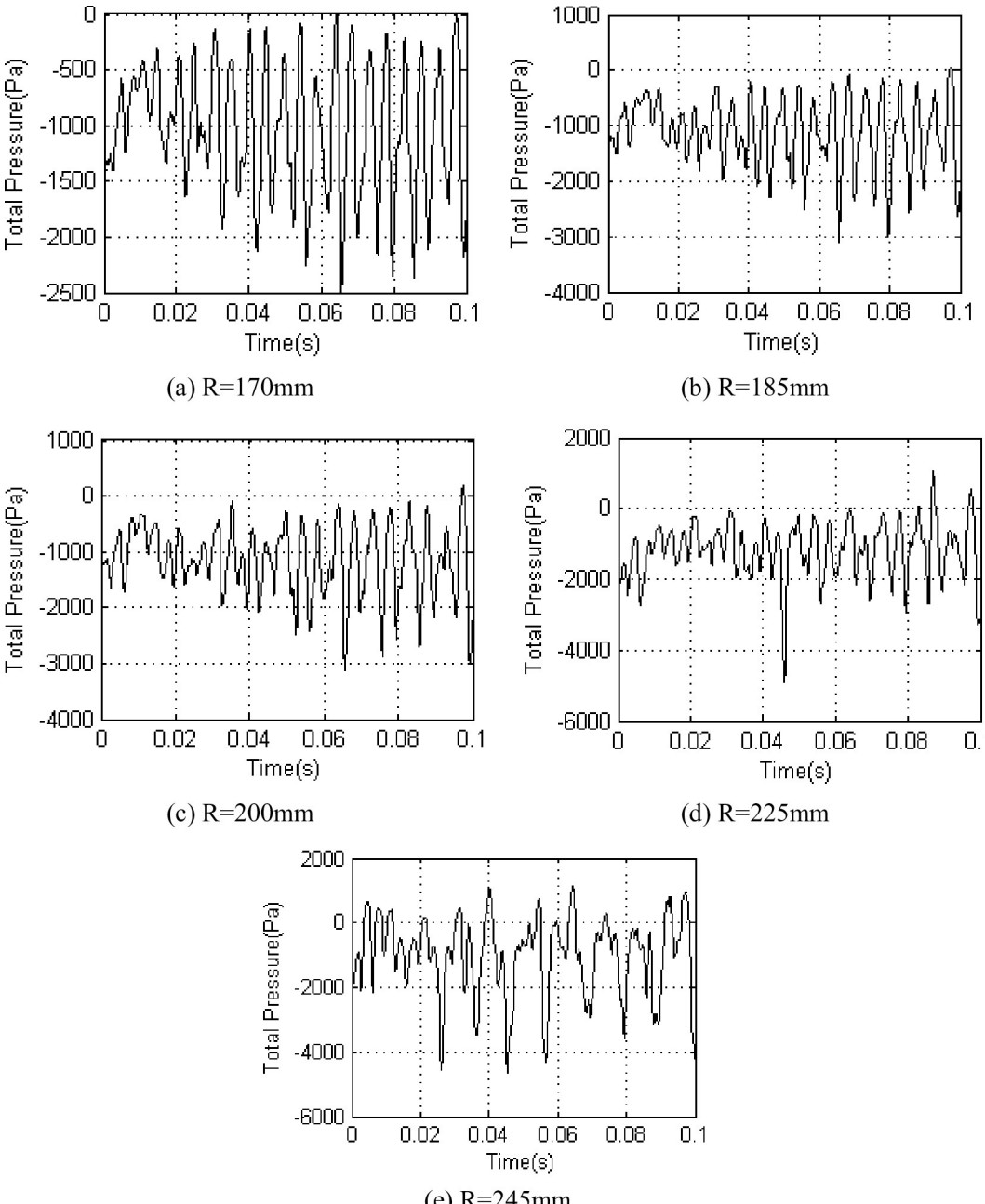

**Fig 4. Interstage pressure pulsation on different radius under Φ = 0.605.**

trailing edges of the blade, the blade and hub wall surface, and the blade tip to handle the turbulence flow [27]. Test calculations were performed at Φ = 0.605 where the performance parameters show a sharp variation. Solutions were assumed to be converged when the residuals of the discretized equation were reduced to 1% of the reference value. Performance parameters and the pressure on the blade surface were chosen as reference variables for testing grid independency, since a primary concern of the present study was to investigate the performance parameters. Although the performance parameters sharply changed at Φ = 0.605 in the performance curve, the grid refinement errors had been estimated within 5%.

The upper section experimental investigations indicated that the interstage airflow pressure pulsation amplitude increased from the blade root to the tip. When Φ = 1 (Fig 3), the airflow pressure exhibited periodic pulsation, and the value fluctuated around -400 Pa at radii of 170, 185 and 200 mm when the pulsation amplitude increased from 450 Pa to 700 Pa. When the value fluctuated around -500 Pa at a radius of 225 mm, the pulsation amplitude was 750 Pa. When the value fluctuated around -1000 Pa at a radius of 245 mm, the pulsation amplitude was 800 Pa. In reduced flow experimental conditions (Fig 4), the periodicity of the airflow pressure pulsation began to weaken from the blade root to the tip, moreover, the airflow pressure pulsation amplitude increased.

### 3.2 Transient stress analysis of second-stage blade

For a whole rotating cycle of the blade in different times, the suction surface total pressure distributions of the second-stage blades are presented in different relative flow coefficients (Figs 5 and 6). The suction surface pressure distribution of the blades exhibited considerable pressure gradient distributions. When Φ = 1, the blade surface pressure do not change significantly over time, and pressure mutations near the blade tip, a negative pressure zone around the blade tip, and a negative pressure extremum appears at the top of the blade leading edge. When Φ = 0.605, the blade surface pressure distribution changes significantly and in a more complex manner over time, a negative pressure zone gradually expands to the blade central zone, and a negative pressure extremum appears at the top of the blade leading edge and blade central zone.

To further study the effect of the variations in the blade pressure distribution on the blades, the stress distribution and deformation diagrams of the second-stage blade were obtained from the fluid-structure coupling flow field in a whole rotating cycle of the blade in different times. We found that the stress distribution of blade was consistent with a typical cantilever beam structure stress distribution, the stress extremum was at the blade root, the blade deformation amplitude was not significant due to the shorter and thicker blades, and the deformation extremum appeared at the top of the blade leading edge. When Φ = 1, the stress distribution and deformation of blade exhibited smaller changes over time, the deformation extremum was at 0.331 mm, and the stress extremum was 103 MPa. When Φ = 0.605, the stress distribution and deformation of blade began to change over time, the deformation extremum ranged from 0.327 to 0.343 mm, and the stress extremum ranged from 101 to 106 MPa. The stress distribution and deformation of blade at the 1T moment for Φ = 0.605 was plotted (Fig 7). We found that under high flow conditions, the airflow pressure pulsation and stress distribution changes were smaller. However, with low flow conditions, the flow field began to degrade, and the airflow pressure pulsations became significant; moreover, the blade root stress began to change.

### 3.3 Wavelet reconstruction analysis of interstage pressure fluctuation

In order to confirm the precision of time delay $\tau$ and embedding dimension $m$, a brief description about the algorithm is given. Let $X = \{x_i(t)\}$, $i = 1,2...,N$, be a part of chaotic time series whose evolution through time is described by a d-dimension dynamical system. Set an initial value for the embedding dimension; that is, let $m = m_0$. Take the time delay $\tau$ as a variable and let it increase by one for each iteration. At each determinated value of $\tau$, reconstruct $X$ into $M = N-(m-1)\tau$ dimensions of vectors $\{x_i\}$, $i = 1,2,...,M$, $x_i = (x_i,x_{i+1},...,x_{i+(m-1)\tau})$ $x_i \in R^m$. Then, calculate the AD of the entire vector space using:

$$S(\tau) = \frac{1}{M}\sum_{i=1}^{M}\sqrt{\sum_{j=1}^{M-1}\left[x_{i+j\tau} - x_i\right]^2} \tag{1}$$

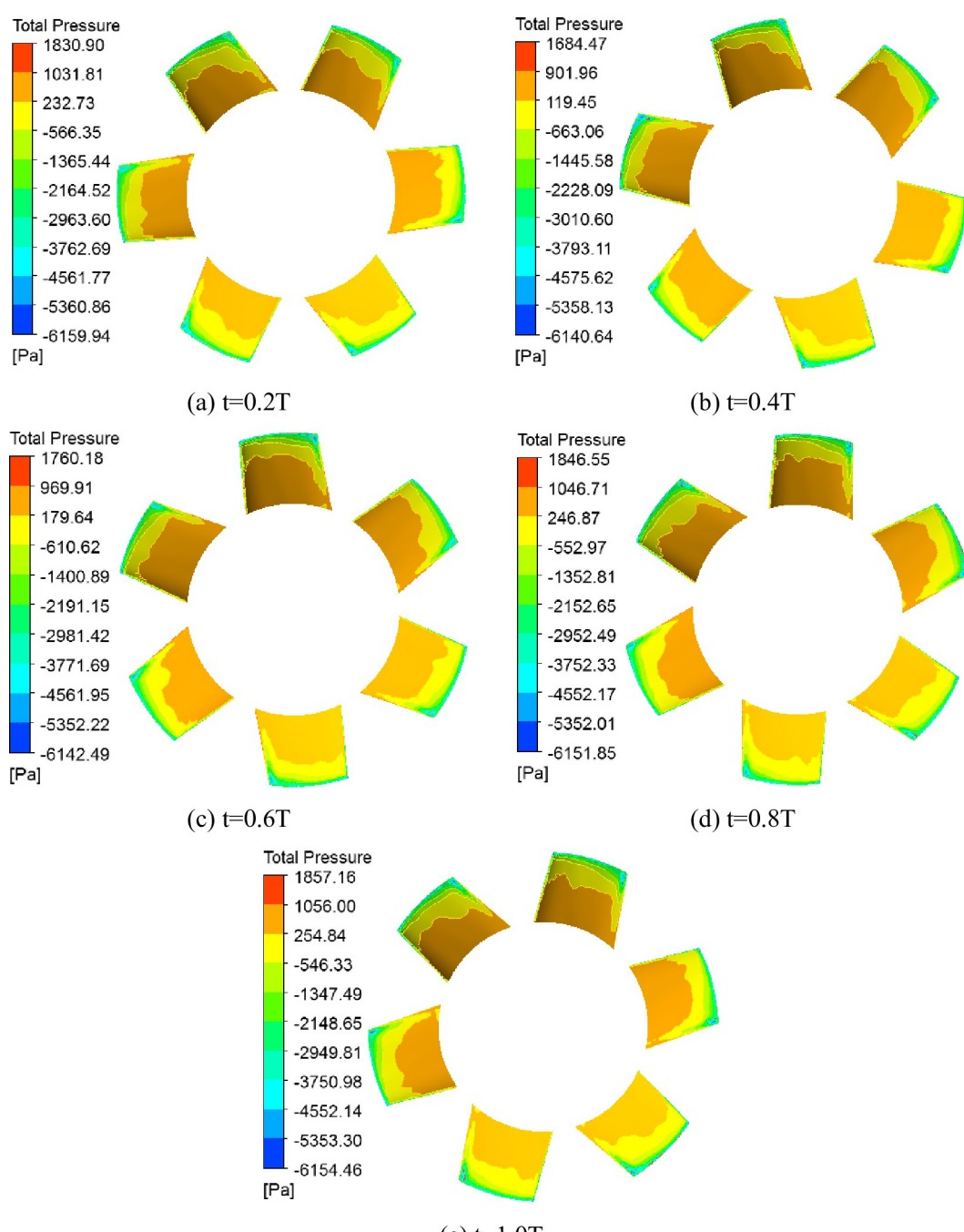

**Fig 5. Total pressure distribution of second-stage blade suction surface under Φ = 1.**

where *M* is the number of data points used for the estimation. As the delay time increases from zero, the reconstructed trajectory expands from the diagonal and $S(\tau)$ increases accordingly until it reaches a plateau. The corresponding value of delay time when $S(\tau)$ gets into saturation is the near-optimum $\tau$ under a certain value of *m*. Through the experimental analysis, if we chose a larger value of $\tau$ and *m*, the correlation between each signal points are reduced, will

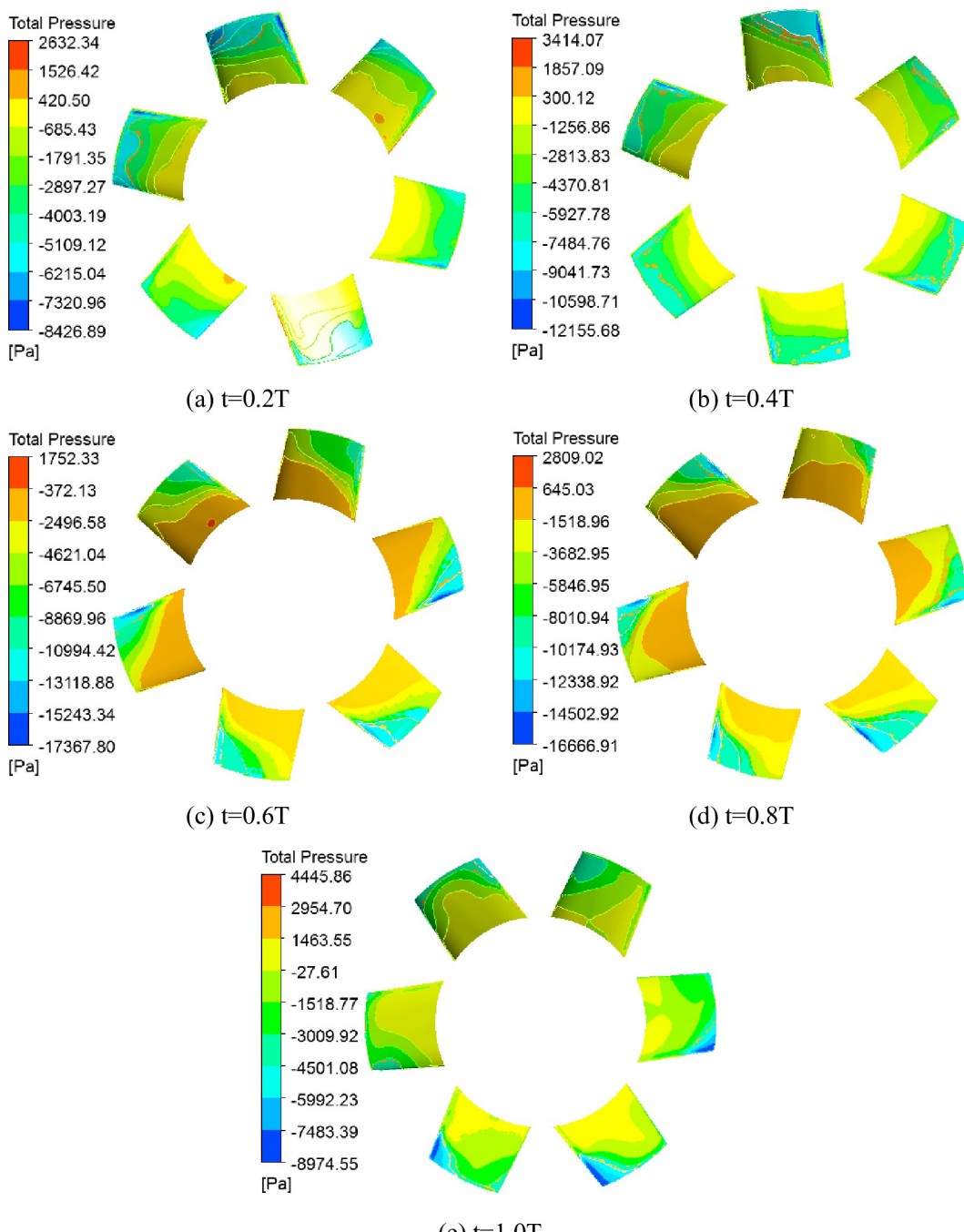

**Fig 6. Total pressure distribution of second-stage blade suction surface under Φ = 0.605.**

cause the strange attractors become complex. So, in this paper, we chose the time delay $\tau = 2$ and embedding dimension $m = 2$.

As previously described, the best performance of the counter-rotating axial fan was $\Phi = 1$ working condition. Otherwise, deviating from the design conditions, the flow condition begun to deteriorate, and flow separation begun to appear. The vortex was caused by flow separation, and under the action of viscous forces and blade coupling vibration, evolved into a smaller

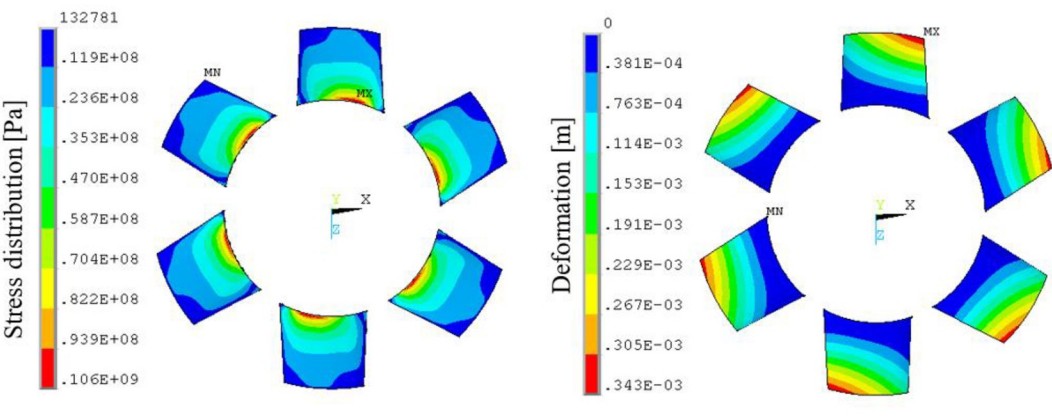

(a) Blade's stress distribution                    (b) Deformation of the blade

**Fig 7. Blade's stress distribution and distortion under t = 1.0T.**

vortex, which caused the airflow pressure pulsations. An FFT was performed at R = 200 mm for the interstage pressure fluctuations (Fig 8).

We found the airflow pressure extremum at 300 ~ 1200 Hz for $\Phi = 1$ and at 0 ~ 1200 Hz for $\Phi = 0.605$. When $\Phi = 0.605$, the airflow exhibited low-frequency pressure pulsation, the pulsation amplitude significantly increased, the frequency signal became complex, and the spectrum changed from a discrete spectrum to a continuous spectrum.

The pressure pulsation signals were processed with wavelet decomposition and reconstruction to further accurately obtain the frequency information of the airflow pressure pulsation. Among all the orthogonal wavelet functions, the dbN wavelet has the shortest compact support under a given order of vanishing moments [28]. The db7 wavelet was chosen as the wavelet function after various experiments and a literature review [29, 30].

Because the db7 wavelet was used for layer 7 wavelet decomposition for the pressure pulsation signal, the detailed signals are d1(1281~2560Hz), d2(641~1280Hz), d3(321~640Hz), d4 (161~320Hz), d5(81~160Hz), d6(41~80Hz), d7(21~40Hz) (Fig 9). We only find the time-domain information of each layer reconstructed signal, therefore, a frequency spectrum

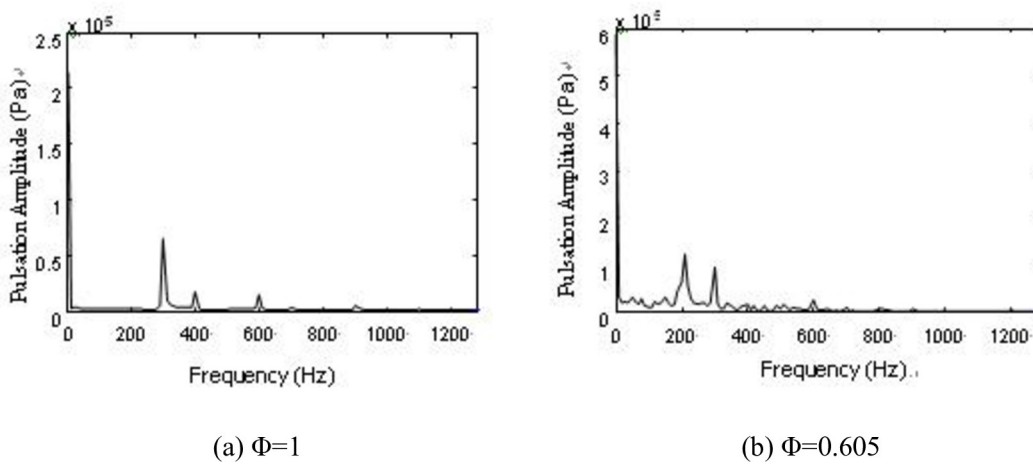

(a) $\Phi=1$                                        (b) $\Phi=0.605$

**Fig 8. FFT of interstage pressure pulsation under R = 200 mm.**

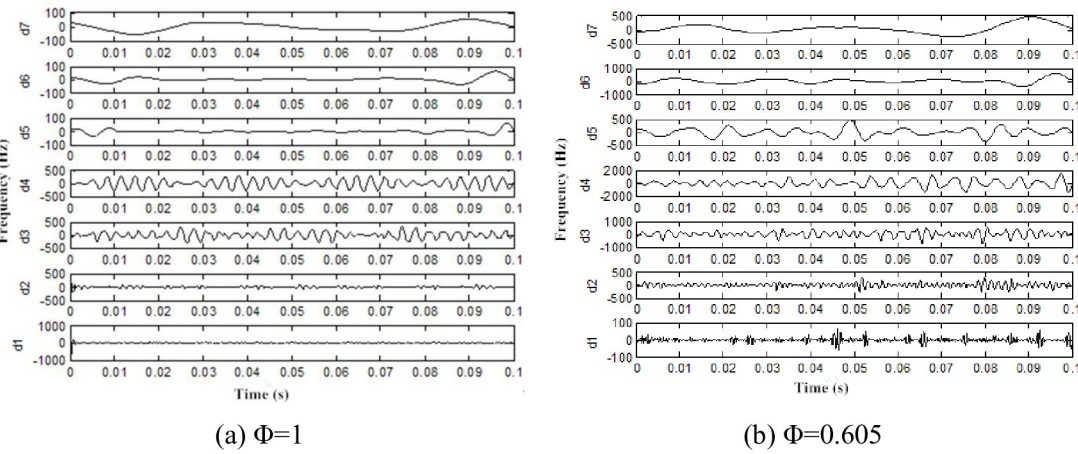

(a) Φ=1                                                                    (b) Φ=0.605

**Fig 9. Restructuring signal of interstage pressure pulsation.**

analysis for each layer reconstruction signal is necessary. Parts of the reconstruction frequency spectrum of signal are presented (Figs 10 and 11).

The frequencies of 300, 400, 600, 690, 780, 880, and 980Hz were found in two working conditions (Fig 10). Moreover, these frequencies were found with significant amplitudes in the frequency spectrum. The rotational fundamental frequency of a two-stage impeller can be calculated according to the formula for rotational frequency [31]:

$$f_r = \frac{nZ}{60} \tag{2}$$

where $n$ is the fan speed (r/min) and $Z$ is the blade number.

The rotational fundamental frequency of the first-stage impeller is 392 Hz, and the second-stage impeller is 294 Hz. Considering the analyzability of the experimental data, frequencies of 400 and 300 Hz are considered as the corresponding experimental data in the allowable error, and the airflow pressure pulsation amplitudes of these frequencies are large. In the same manner, frequencies of 600 and 880 Hz are considered as the rotational fundamental frequencies of the second-stage impeller and the second and third harmonic waves, 780 Hz is considered as the rotational fundamental frequency of the first-stage impeller and twice the harmonic frequency, and 690 and 880 Hz are considered as the superposition rotational fundamental frequency of the two-stage impellers. The amplitude of 690 Hz is the most significant (Fig 10). At

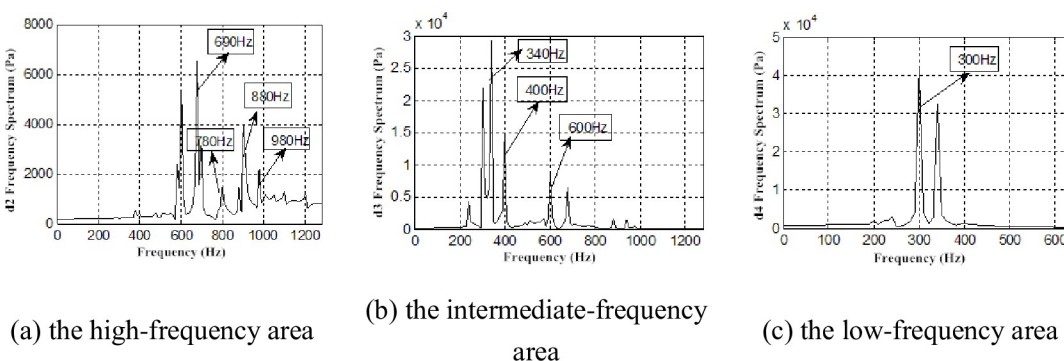

(a) the high-frequency area          (b) the intermediate-frequency area          (c) the low-frequency area

**Fig 10. Frequency spectrum of interstage pressure pulsation restructuring signal under Φ = 1.**

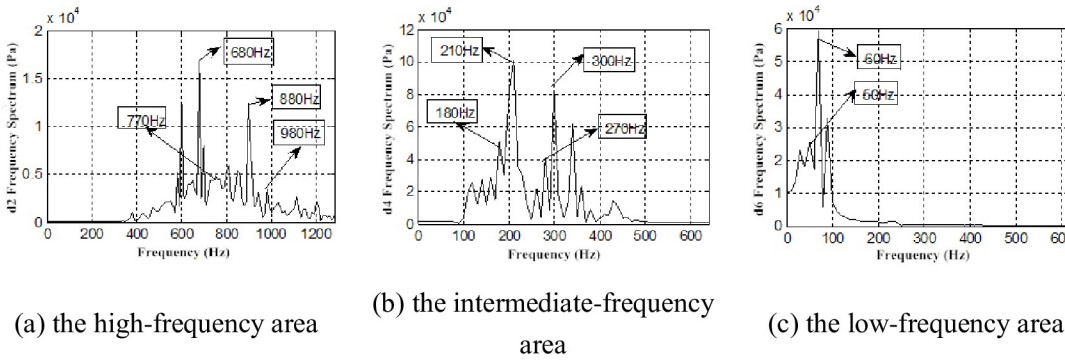

| (a) the high-frequency area | (b) the intermediate-frequency area | (c) the low-frequency area |

**Fig 11. Frequency spectrum of interstage pressure pulsation restructuring signal under Φ = 0.605.**

Φ = 0.605, there are the frequencies 680 and 770 Hz phase lag phenomena caused by airflow decrease, viscous diffusion and the flow field deterioration (Fig 11).

Compared with the frequency spectrum of the interstage pressure pulsation and the reconstructed signal under the two working conditions, the frequency spectrum of each layer reconstructed signal exhibits a discrete spectrum with Φ = 1 and a wide-band continuous spectrum with Φ = 0.605. The amplitudes of the frequencies of 400 and 300Hz remained larger (Fig 10). We also find low-frequency airflow pulsations of 30, 60, 90, and 150 Hz (Fig 11). Rotating stall phenomena are observed in the coupling flow field at Φ = 0.605, and the frequency of the rotating stall group is 0.2 ~ 0.8 times. As a general rule, the rotational frequency is 2/3. The frequency of 30Hz is approximately 2/3 frequency of 49Hz. The frequencies of 60, 90, 150, 180, and 210Hz are two, three, five, six, seven times the harmonic frequency of the rotating stall group. This proves the existence of the rotating stall phenomenon as well. Therefore, the continuous spectrum characteristics of the decreased flow results from the low-frequency airflow pulsation caused by rotating stall, vortex movement and breakdown.

## 3.4 Fluid-structure coupling vibration signal analysis of second-stage blade

The change between the Y direction and X direction displacements in the suction leading edge point surface of the second-stage blade along with time are plotted (Figs 12 and 13). The blade

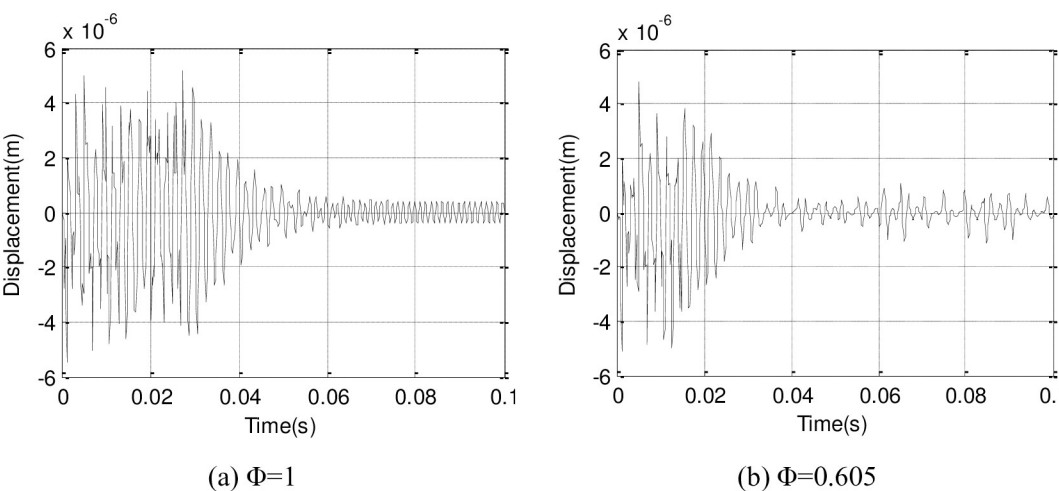

| (a) Φ=1 | (b) Φ=0.605 |

**Fig 12. Y direction displacement response of second-stage blade.**

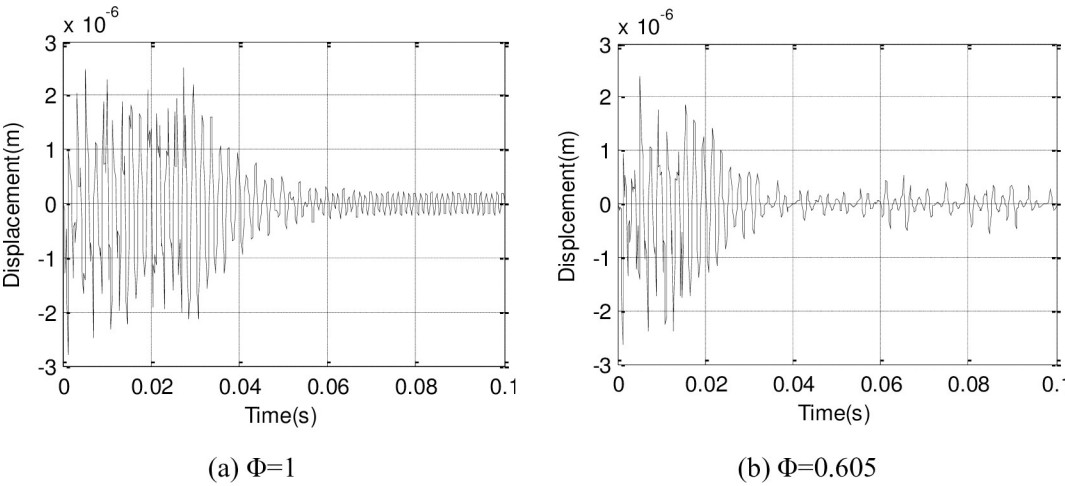

(a) Φ=1 (b) Φ=0.605

**Fig 13. X direction displacement response of second-stage blade.**

displacement response tends to convergence and becomes steady after 0.05s, and the amplitude is larger in the reduced flow.

A frequency spectrum analysis is conducted for the displacement responses (Figs 14 and 15), and the 500Hz measurement approximates the 494Hz of the first-order natural frequency of second-stage blade. Therefore, the fluid-structure coupling flow field is mainly excited by the first-order vibration of the second-stage blade.

To further accurately measure frequency information of the blade vibration response, the db7 wavelet is used to perform layer 7 wavelet decomposition on the vibration signals, and the detailed signal of each layer is measured with a frequency spectrum analysis after reconstruction. Parts of the frequency spectrum of the reconstructed signal are presented (Figs 16–19).

Besides, the first-order vibration frequency of second-stage blade, we also find the vibration frequency to be consistent with the airflow pulsation frequency (290, 680, 780, 880 Hz) (Figs 16–19), which is also consistent with previous research [32]. We also demonstrate that the two-stage blade cut airflow to make flow field change, thus, the airflow pulsation is produced

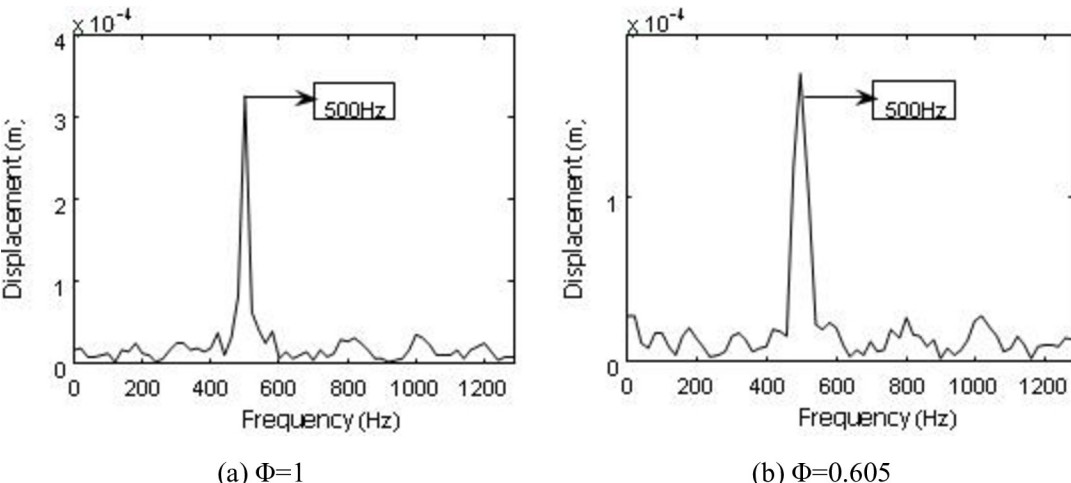

(a) Φ=1 (b) Φ=0.605

**Fig 14. FFT of the second stage blade Y direction displacement response.**

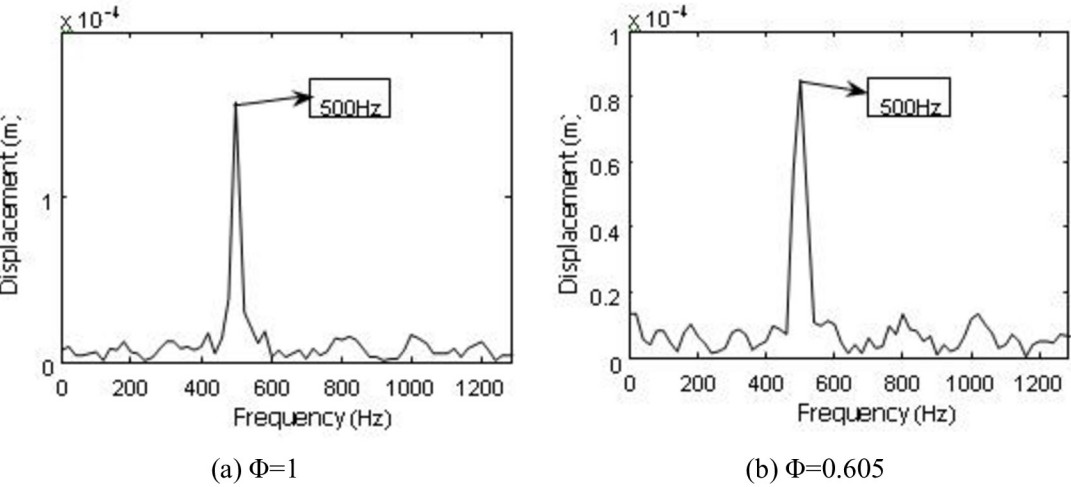

(a) Φ=1                                    (b) Φ=0.605

**Fig 15. FFT of the second stage blade X direction displacement response.**

to cause the blade deformation and the same frequency vibration. On the contrary, the blade deformation also affects the flow field.

In addition, the low-frequency zone vibration presents higher activity under low flow conditions, and the frequencies 30, 60, 90, 150, 180 and 270 Hz are found. It can be deduced that these frequencies are the vortex vibration frequencies caused by rotating stall, vortex movement and breakdown.

The vortex vibration frequency can be calculated by the empirical formula [33]

$$f_i = S_r w i / L \tag{3}$$

where $S_r$ is the Strouhal number, $S_r = 0.14 \sim 0.2$, $w$ is the relative speed between the airflow and blade, $L$ is the projection of the blade positive surface width perpendicular to the speed plane, and $i$ is the harmonic number.

The vortex vibration frequency is approximately 20~120Hz [22]. The blade vortex vibration is caused by flow field deterioration with rotating stall, vortex movement and breakdown. Although the vortex vibration amplitude is small, it could reduce the fan's efficiency.

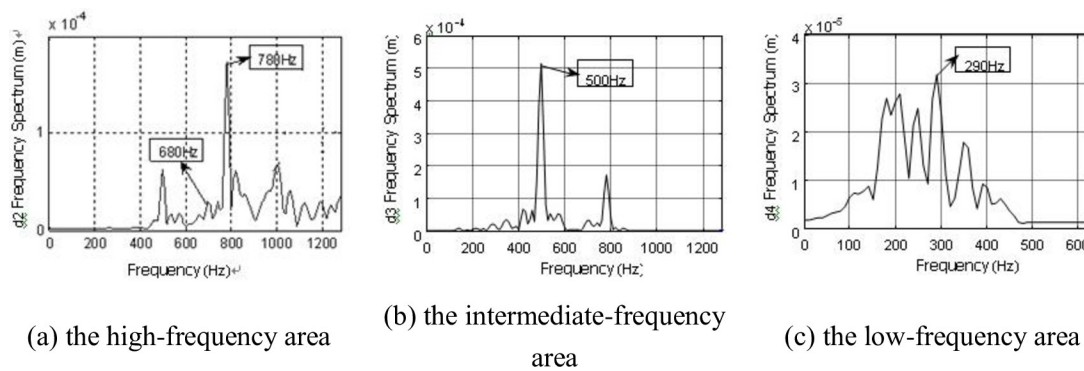

(a) the high-frequency area    (b) the intermediate-frequency area    (c) the low-frequency area

**Fig 16. Frequency spectrum of Y direction displacement response restructuring signal under Φ = 1.**

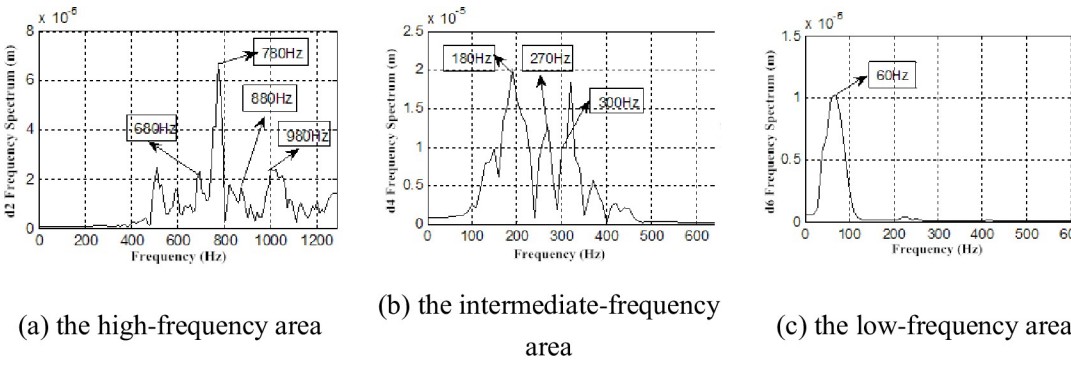

(a) the high-frequency area

(b) the intermediate-frequency area

(c) the low-frequency area

**Fig 17. Frequency spectrum of Y direction displacement response restructuring signal under Φ = 0.605.**

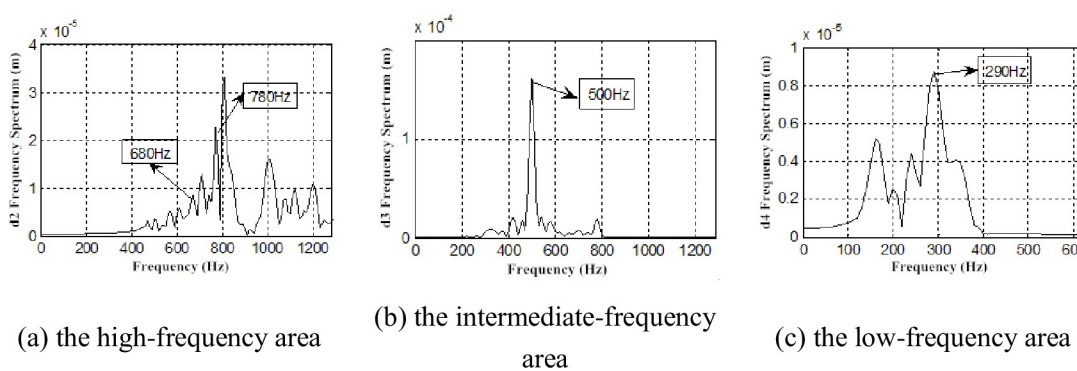

(a) the high-frequency area

(b) the intermediate-frequency area

(c) the low-frequency area

**Fig 18. Frequency spectrum of X direction displacement response restructuring signal under Φ = 1.**

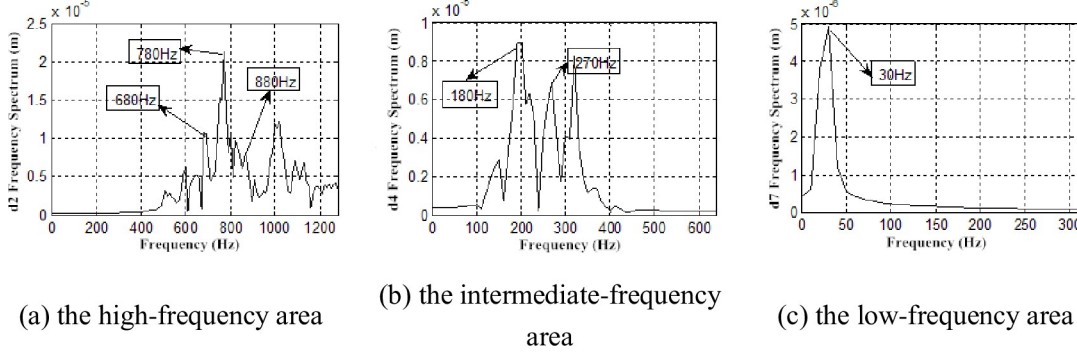

(a) the high-frequency area

(b) the intermediate-frequency area

(c) the low-frequency area

**Fig 19. Frequency spectrum of X direction displacement response restructuring signal under Φ = 0.605.**

## 4. Conclusions

In this work, the signal processing method of wavelet decomposition and reconstruction in time-frequency analysis process are presented experimentally and numerically in the contra-rotating fan, and the following conclusions can be received that:

1. The instability of the flow field initially appears around the blade tip, and the flow regime of the blade tip is more complex than that of the blade root, which is mainly caused by the

existence of a blade tip leakage flow. The leakage flow gradually develops from the blade leading edge, and the strong airflow pressure pulsation of blade tip due to the periodic blade vibration, and then, the obvious generated vortex disrupted the periodicity of the pressure pulsation.

2. In the rated flow working condition, the internal flow of the counter-rotating axial fan is steady, the flow field pressure pulsation is smooth and steady. In addition, the stress and deformation of the second-stage blade are steady, and the frequency spectrum of the interstage pressure pulsation exhibit a discrete spectrum. When $\Phi = 0.605$, the internal flow field of the counter-rotating axial fan begin to deteriorate, and the phenomenon of the inlet vortex, flow channel vortex, boundary layer separation and rotating stall begin to present themselves.

3. The stress and deformation of the second-stage blade exhibits slight changes, and the frequency spectrum of the interstage pressure pulsation exhibits a continuous spectrum. The results indicate that the fluid-structure coupling interaction field in the interstage of the counter-rotating axial fans mainly excites the first-order vibration of the second-stage blade. At the same time, the consistency between the pulsation frequency and the vibrational frequency of the airflow reflects the good coupling property.

## Acknowledgments

We thank the associate editor and anonymous reviewers for their helpful comments.

## Author Contributions

**Formal analysis:** Chen Dai, Guojie Zhang.

**Investigation:** Guojie Zhang.

**Methodology:** Xinzhe Zhang.

**Resources:** Sławomir Dykas.

**Software:** Xinzhe Zhang, Sławomir Dykas.

**Supervision:** Guojie Zhang.

**Validation:** Xinzhe Zhang, Sławomir Dykas.

**Writing – original draft:** Guojie Zhang.

**Writing – review & editing:** Chen Dai, Zunlong Jin.

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
