## [Decision Letter · Decision Letter 0]

9 Nov 2020

PONE-D-20-29616

Coupling Analysis of Contra-Rotating Fan Interstage Pressure Pulsation and Blade Vibration based on Wavelet Reconstruction

PLOS ONE

Dear Dr. Zhang,

Thank you for submitting your manuscript to PLOS ONE. After careful consideration, we feel that it has merit but does not fully meet PLOS ONE’s publication criteria as it currently stands. Therefore, we invite you to submit a revised version of the manuscript that addresses the points raised during the review process.

We look forward to receiving your revised manuscript.

Kind regards,

Hongbing Ding, Ph.D.

Academic Editor

PLOS ONE

Additional Editor Comments:

Thank you for submitting your manuscript to PLOS ONE. The reviewers recommend reconsideration of your paper following minor revision. I invite you to resubmit your manuscript after addressing all reviewer comments.

Journal Requirements:

We note that one or more of the authors are employed by a commercial company: Censtar Science & Technology Corp., Ltd.

5.1. Please provide an amended Funding Statement declaring this commercial affiliation, as well as a statement regarding the Role of Funders in your study. If the funding organization did not play a role in the study design, data collection and analysis, decision to publish, or preparation of the manuscript and only provided financial support in the form of authors' salaries and/or research materials, please review your statements relating to the author contributions, and ensure you have specifically and accurately indicated the role(s) that these authors had in your study. You can update author roles in the Author Contributions section of the online submission form.

5.2. Please also provide an updated Competing Interests Statement declaring this commercial affiliation along with any other relevant declarations relating to employment, consultancy, patents, products in development, or marketed products, etc.  

Please respond by return email with an updated Funding Statement and Competing Interests Statement and we will change the online submission form on your behalf.

Reviewers' comments:

Reviewer's Responses to Questions

**Comments to the Author**

1. Is the manuscript technically sound, and do the data support the conclusions?

Reviewer #1: Yes

Reviewer #2: Yes

2. Has the statistical analysis been performed appropriately and rigorously? 

Reviewer #1: Yes

Reviewer #2: Yes

3. Have the authors made all data underlying the findings in their manuscript fully available?

Reviewer #1: Yes

Reviewer #2: Yes

4. Is the manuscript presented in an intelligible fashion and written in standard English?

Reviewer #1: Yes

Reviewer #2: Yes

5. Review Comments to the Author

Reviewer #1: the pressure pulsation of a contra-rotating fan is analyzed using experimental and numerical methods, including wavelet reconstruction method, which is meaningful for fan design and optimization. Although this manuscript is written well, there are some suggestions:

1. the English need be polished carefully.

2. the reference is old, some fresh references need be added.

3. the numerical boundary need be explained in detailed.

4.the conclusions need be re-written carefully to show the work’s novelty.

Reviewer #2: The fluid-structure coupling interaction field in the interstage of counter-rotating axial fans mainly excites the first-order vibration of the second-stage blade. The fluid-structure coupling interaction field in the interstage of counter-rotating axial fans mainly excites the first-order vibration of the second-stage blade. This research is very interesting and has some engineering application value.It is suggested to accept after modification.

1)The quality of the graph needs to be improved, whether it is compressed or not is unknown.

2)The format of references is not uniform, please modify according to the standard template.

3)Some sentences have grammatical errors, so it is recommended to polish the professional language.

6. PLOS authors have the option to publish the peer review history of their article (what does this mean?). If published, this will include your full peer review and any attached files.

Reviewer #1: No

Reviewer #2: No

---

## [Author Response · Author response to Decision Letter 0]

5 Jan 2021

Dear editor and reviewers:

Thank you for your letter and the reviewers’ comments on our manuscript entitled ‘Coupling Analysis of Contra-Rotating Fan Interstage Pressure Pulsation and Blade Vibration based on Wavelet Reconstruction’ (the manuscript number is PONE-D-20-29616). Those comments are very helpful for revising and improving our paper, as well as the important guiding significance to other’s research. We have studied the comments carefully and made corrections, we hope the manuscript can be accepted for publication. The main corrections are in the manuscript and the responses to the reviewers’ comments are as follows (the replies are highlighted in red).

Comments from the editors and reviewers:

Comments from the reviewers:

Reviewer #1: the pressure pulsation of a contra-rotating fan is analyzed using experimental and numerical methods, including wavelet reconstruction method, which is meaningful for fan design and optimization. Although this manuscript is written well, there are some suggestions:

1. the English need be polished carefully.

Response to the Reviewer: Thank you for your valuable and careful comments. We have revised the paper, and the English is polished carefully. I hope it can meet your requirements. Thank you again!

2. the reference is old, some fresh references need be added.

Response to the Reviewer: Thank you for your valuable and careful comments. We have added lots of last references. I hope it can meet your requirements. Thank you again!

3. the numerical boundary need be explained in detailed.

Response to the Reviewer: Thank you for your valuable and careful comments. We have revised this section, and the detailed boundary conditions are added. I hope it can meet your requirements. Thank you again for your careful comments.

4.the conclusions need be re-written carefully to show the work’s novelty.

Response to the Reviewer: Thank you for your valuable and careful comments. The conclusions have been revised carefully, I hope it can show the work’s novelty and meet your requirements. Thank you again for your review again!

Reviewer #2: The fluid-structure coupling interaction field in the interstage of counter-rotating axial fans mainly excites the first-order vibration of the second-stage blade. The fluid-structure coupling interaction field in the interstage of counter-rotating axial fans mainly excites the first-order vibration of the second-stage blade. This research is very interesting and has some engineering application value. It is suggested to accept after modification.

1)The quality of the graph needs to be improved, whether it is compressed or not is unknown.

Response to the Reviewer: Thank you for your valuable and careful comments. The graph is exported by the experimental equipment. Thank you for your professional comments again! 

2)The format of references is not uniform, please modify according to the standard template.

Response to the Reviewer: Thank you for your careful comments. We have revised the reference format following the journal requirement. Thank you again for your carefully comments again!

3)Some sentences have grammatical errors, so it is recommended to polish the professional language.

Response to the Reviewer: Thank you for your careful comments. We have revised the paper, and the English is polished carefully. I hope it can meet your requirements. Thank you again!

---

## [Editor Report · Decision Letter 1]

12 Jan 2021

Coupling Analysis of Contra-Rotating Fan Interstage Pressure Pulsation and Blade Vibration based on Wavelet Reconstruction

PONE-D-20-29616R1

Dear Dr. Zhang,

We’re pleased to inform you that your manuscript has been judged scientifically suitable for publication and will be formally accepted for publication once it meets all outstanding technical requirements.

Kind regards,

Hongbing Ding, Ph.D.

Academic Editor

PLOS ONE

Additional Editor Comments (optional):

The authors have done a good job in revising the manuscript. Now it can be accepted for publication in PLOS ONE.
---

## [Editor Report · Acceptance letter]

29 Jan 2021

PONE-D-20-29616R1 

Coupling Analysis of Contra-Rotating Fan Interstage Pressure Pulsation and Blade Vibration based on Wavelet Reconstruction 

Dear Dr. Zhang:

I'm pleased to inform you that your manuscript has been deemed suitable for publication in PLOS ONE. Congratulations! Your manuscript is now with our production department. 

Kind regards, 

on behalf of

Professor Hongbing Ding 

Academic Editor

PLOS ONE